

# Integrable deformations of superintegrable quantum circuits

**Tamás Gombor**[⋆] **and Balázs Pozsgay**[†]

MTA-ELTE "Momentum" Integrable Quantum Dynamics Research Group,
Department of Theoretical Physics, Eötvös Loránd University,
Pázmány Péter stny. 1A, Budapest 1117, Hungary

⋆ gombor.tamas@wigner.hu , † pozsgay.balazs@ttk.elte.hu

## Abstract

Superintegrable models are very special dynamical systems: they possess more conservation laws than what is necessary for integrability. This severely constrains their dynamical processes, and it often leads to their exact solvability, even in non-equilibrium situations. In this paper we consider special Hamiltonian deformations of superintegrable quantum circuits. The deformations break superintegrability, but they preserve integrability. We focus on a selection of concrete models and show that for each model there is an (at least) one parameter family of integrable deformations. Our most interesting example is the so-called Rule54 model. We show that the model is compatible with a one parameter family of Yang-Baxter integrable spin chains with six-site interaction. Therefore, the Rule54 model does not have a unique integrability structure, instead it lies at the intersection of a family of quantum integrable models.

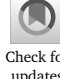

# 1 Introduction

One dimensional integrable models are special dynamical systems, which allow for an exact solution. This means that it is possible to compute certain physical quantities, in equilibrium or out-of-equilibrium situations. A common characteristic of integrable models (both for classical and quantum mechanical systems) is the existence of a large set of conservation laws [1, 2]. These constrain the dynamics, and they distinguish the integrable models from the chaotic systems, which only have a handful of conservation laws, following from global symmetries.

In classical mechanics a system with $n$ degrees of freedom (having a $2n$ dimensional phase space) is integrable if it has $n$ algebraically independent charges (functions on the phase space which commute with each other under the Poisson bracket). Superintegrable systems are even more special models, which have more than $n$ conserved charges, see for example [3]. Perhaps the most famous example is the Kepler problem, where the Laplace-Runge-Lenz vector provides an extra conservation law, facilitating the algebraic determination of the orbits.[1]

The notion of integrability is less clear in quantum many body models, but the presence of a large set of conserved charges is regarded as common characteristics of such systems.[2] One might then wonder what does superintegrability mean for quantum many body physics. The natural answer is that a superintegrable model has more charges than necessary for integrability, and its dynamics is even more constrained. However, this is just a vague characterisation, and later in the main text we provide a more precise definition.

The motivation to consider superintegrable quantum models comes from non-equilibrium physics. In the last decade considerable efforts were spent to study the non-equilibrium behaviour of integrable models (thermalization, transport properties, etc). In non-equilibrium situations one needs to deal with a large number eigenstates, and in a standard integrable model (such as the Heisenberg spin chains or the 1D Bose gas) this becomes a difficult task for both analytic and numerical approaches. This motivated researchers to consider special models with even simpler dynamics, and some of these models turn out to be superintegrable.

Perhaps the most famous example is the so-called Rule54 model [5], which is often called the simplest interacting integrable model. It is a cellular automaton, which has both a classical and a quantum formulation, and it has been in the forefront of research in the last 5 years, see the review article [6]. The model supports right moving and left moving quasiparticles (solitons) which propagate with constant speed $\pm 1$, and which scatter on each other, suffering a non-zero scattering displacement. The resulting dynamics is simple enough so that certain non-equilibrium properties of the model could be computed analytically, including equilibration and transport phenomena and also entanglement production [7–10] (see also [11–13]). The Rule54 model is superintegrable: the classical formulation has an exponential number of local conservation laws [6] in the number of spins, and the eigenvalue spectrum of the associated Floquet operator is exponentially degenerate [12].

Perhaps surprisingly, despite the large number of results obtained for this model the actual algebraic origin of its integrability has not yet been understood. An attempt was made in [14] to embed the Rule54 model into the canonical framework of Yang-Baxter integrability [2, 15] but it was shown in [16] that the approach of [14] does not yield new conserved charges, and

---

[1]For completeness, we summarize the superintegrability of the Kepler problem, which is most easily treated in the relative coordinate. This way it becomes a one body problem in the plane, and it has $n = 2$ degrees of freedom. There are $n = 2$ standard conservation laws: the energy and the angular momentum. The Runge-Lenz vector is conserved as a vector, but its magnitude is a function of the energy and the angular momentum. However, its direction is a new conserved quantity, making the problem superintegrable.

[2]The crucial distinction between the classical and the quantum mechanical cases is that in quantum mechanics it is not enough to specify the number of conserved operators, because every projector $P_\psi = |\psi\rangle\langle\psi|$ to an eigenstate $|\psi\rangle$ is conserved. Therefore, one needs specific statements/requirements about the structure of the operators, representing the conserved quantities. For an extensive discussion of this problem we refer to [4].

it only reproduces a few known ones. A hint towards a potential Yang-Baxter structure was provided in [12], where a six-site quantum charge was found which commutes with discrete time update step (Floquet operator) of the model. This six-site charge was used to deform the model in a way which preserves its integrability, but which destroys the superintegrability. The presence of this extra six-site Hamiltonian gives dispersion to the quasiparticles, which is enough to lift the exponential degeneracies and break most of the conservation laws of the original superintegrable model. However, the algebraic integrability of the resulting model was not clarified in [12]. Interestingly, the "space-like dynamics" of the model also involves operators with longer range: A deterministic "space-like" evolution with five-site operators was formulated in [17].

In this work we revisit the Rule54 model and other superintegrable quantum cellular automata. Following [12] we consider the problem of Hamiltonian deformations of these models. Our goal is to deform these models away from superintegrability, but preserving their integrability. A reformulation of the problem is the following: Our goal is to find integrable Hamiltonians with well defined Yang-Baxter structures which commute with the time evolution of the selected superintegrable quantum circuits.

We find a somewhat unexpected phenomenon: The integrable deformation of the models we consider is not unique. In fact, there appears to be an (at least) one parameter family of integrable Hamiltonians which commute with the superintegrable cellular automata. This means that the Rule54 model and a few similar models that we treat do not specify a unique integrable structure, instead they lie at an intersection of a continuous family of integrable models. To our best knowledge this phenomenon has not yet been noticed in the literature.

In Section 2 we set the stage: We introduce the framework for continuous and discrete time evolution in one dimensional quantum spin chains. In Sections 3-5 we discuss the concrete examples for the integrable deformations. The three examples that we treat have increasing complexity. First, in Section 3 we treat the permutation or SWAP circuit, which has a completely trivial dynamics, allowing for practically infinite possibilities for integrable deformations. Second, in Section 4 we add non-trivial phase factors to the SWAP circuit. The resulting model is dual-unitary [18] and integrable; its non-equilibrium properties were treated recently in [19]. Here we consider its Hamiltonian deformations. Finally, in Section 5 we treat the Rule54 model, which is actually our main and most involved example. We discuss our findings in Section 6, and in the Appendix A we provide details about the Yang-Baxter integrability of the spin chains with the six-site charges, which are related to the Rule54 model.

## 2 Superintegrable quantum circuits

We consider quantum spin chains, with both continuous (i.e., Hamiltonian) and discrete time evolution (i.e., Floquet). The local Hilbert spaces are chosen as $\mathbb{C}^d$ with some $d \geq 2$, and the full Hilbert space is the $L$-fold tensor product, where $L$ is the length of the spin chain. For simplicity we consider periodic boundary conditions, and $L$ is assumed to be an even number.

Our main focus is on quantum circuits (also called quantum block cellular automata), where discrete time evolution is constructed from the action of local quantum update steps, which are performed by local unitary operations. We build circuits of the "brickwork type" [20, 21]. Let $|\Psi(t)\rangle$ be the state of the system at time $t \in \mathbb{Z}$, then the update is performed as

$$|\Psi(t+1)\rangle = \begin{cases} \mathcal{V}_1|\Psi(t)\rangle, & \text{if } t \text{ is even,} \\ \mathcal{V}_2|\Psi(t)\rangle, & \text{if } t \text{ is odd,} \end{cases} \tag{1}$$

where $\mathcal{V}_1$ and $\mathcal{V}_2$ are constructed from a product of mutually commuting local unitary gates.

In the most often used case we consider unitary two-site gate $U$, and

$$\begin{aligned}
\mathcal{V}_1 &= U_{L-1,L} \ldots U_{34} U_{12}, \\
\mathcal{V}_2 &= U_{L,1} \ldots U_{45} U_{23},
\end{aligned} \tag{2}$$

where the two-site gates $U_{j,k}$ are the same operators and the subscripts denote the sites where they act. The product $\mathcal{V} = \mathcal{V}_2 \mathcal{V}_1$ is called the Floquet operator and the structure of $\mathcal{V}$ defines the notion of a brickwork circuit. Time evolution generated by $\mathcal{V}$ has spatial and temporal periodicity equal to 2.

Such circuits can show a variety of physical behaviour, ranging from chaotic to integrable (or more exotic ones: localization, fragmentation / shattering, scars, etc.), including superintegrable cases. We say that a circuit of this type is integrable, if there exists a set of charges $\{Q_\alpha\}$ with the following requirements:

- Each operator is extensive with a local operator density, meaning that $Q_\alpha = \sum_{j=1}^{L/2} q_\alpha(2j)$, where $q_\alpha$ is an operator spanning a finite number of sites, positioned at site $2j$. Note that the spatial periodicity of the charges is 2, in correspondence with the geometry.

- Each charge commutes with the Floquet operator $\mathcal{V}$.

- The charges also commute with each other.

In a standard integrable model with short range interactions the number of available charges grows typically linearly with the volume or the range[3] of the operator density of the charges. For example, in the Heisenberg spin chains (and many other models constructed from local Lax operators [15]) there is precisely one new charge for every range $r$. This is to be contrasted with the behaviour of superintegrable models.

Superintegrability is a concept which has its origins in classical integrability. There a model is called superintegrable, if it has more conservation laws than the degrees of freedom (more than $n$ conservation laws in a $2n$ dimensional phase space). In such models it is often not necessary to actually solve the time evolution, and in many cases information can be obtained simply by algebraic means. In contrast, the notion of superintegrability is less clear in quantum mechanical many body models. In this paper we adopt the following definition:

*A quantum circuit (a spin chain with discrete time evolution constructed from local update rules) is called superintegrable, if it possesses a large set of extensive operators commuting with the time evolution, such that the number of charges with a given range $r$ grows exponentially with $r$.*

Note that we did not require that all charges should commute with each other, we are just concerned with the commutation with the time evolution, which implies conservation of the mean values. This is analogous to the situation in classical mechanics: It is known that if a system has $n$ degrees of freedom, then the maximal number of Poission commuting and algebraically independent functions is $n$ [3]. If there are additional conserved quantities, then they can not commute with all the other charges. However, the conservation of the extra charges will already pose very strong constraints for the dynamics of the superintegrable models, both in the classical and in the quantum mechanical setting. We also note that the exponential growth does not mean that the models are trivial: the growth is typically slower than the growth of the Hilbert spaces.

There is a further common characteristic of superintegrable models, which sets them apart from both the chaotic and the standard integrable systems. In standard integrable models the

---

[3] In this paper we define the range of an operator as the size of its support.

spectrum is such that there are typically no extra degeneracies on the top of those enforced by global symmetries, while the level spacing statistics follows the Poisson statistics. In contrast, in superintegrable models one typically finds exponentially large degeneracies, even in the middle of the spectrum. However, some care needs to be taken at this point. In the case of the quantum circuits the model is defined by the Floquet operators $\mathcal{V}$, which are unitary operators. For a given model let $\lambda_j = e^{i\varepsilon_j}$ denote the eigenvalues of $\mathcal{V}$. The $\varepsilon_j \in \mathbb{R}$ are called "quasi-energies" and they are defined only modulo $2\pi$. Whereas the concept of a ground state is missing in such models, the level spacing statistics can be defined for the $\varepsilon_j$, and one finds the same distinctions between chaotic, integrable and super-integrable circuits.

Having discussed the characteristics of superintegrability let us now turn to the construction of such models. As far as we know, there is no general technique to construct superintegrable circuits, instead there are a few known mechanisms for superintegrability.

For example, one of the possibilities is the presence of gliders. We say that a local operator $\mathcal{O}(x)$ is a glider, if its time evolution (in Heisenberg picture) is a mere translation to the left or to the right. More concretely, the condition for $\mathcal{O}(x)$ to be a glider is

$$\mathcal{V}^\dagger \mathcal{O}(x)\mathcal{V} = \mathcal{O}(x \pm 2), \tag{3}$$

where the two signs describe right- and left-moving gliders, respectively.

Gliders form a closed operator algebra under addition and multiplication: Any product of gliders moving in the same direction is also a glider [22]. This implies that if a model has at least one non-trivial glider, then it has infinitely many, and the spatial sums

$$\sum_{x=\text{odd/even}} \mathcal{O}(x), \tag{4}$$

are conserved during time evolution. Here the summation runs over the odd or even sites of the lattice, depending on the glider in question.

It follows from the above, that whenever there is at least one glider in the model, then the number of linearly independent extensive charges grows exponentially with the range of the charge.

Gliders can be constructed in special cases, when the two-site unitary operator $U$ satisfies the braid relation (spectral parameter independent Yang-Baxter equation) [23]. Other examples can be found in the so-called dual unitary circuits [18], where it is known that all conserved charges come from gliders [22]. Non-trivial examples for dual unitary circuits with gliders were found in [24], including models where the shortest glider spans three- or even five-sites. The integrability properties of these models are not yet understood.

## Hamiltonian deformations

For a given integrable or superintegrable Floquet operator $\mathcal{V}$ let $H$ be a conserved charge. We say that the time evolution operator

$$\mathcal{V}(\lambda) = e^{-iH\lambda}\mathcal{V}, \tag{5}$$

is a Hamiltonian deformation of the quantum circuit, where now $\lambda \in \mathbb{R}$ is a perturbation parameter [12]. Such a deformation was introduced in [12] for the Rule54 model. The advantage of introducing the deformation is that it adds dispersion to the particle propagation, and it lifts the large degeneracies of the original model.

In this paper we show that if the original model is superintegrable, then often there are different families of possible Hamiltonian deformations, which actually lead to different integrable models. To be more precise, we will show that in certain cases we can find an (at least)

one parameter family of integrable models which commute with a given Floquet operator $\mathcal{V}$. This means that there exists a set of charges $\{Q_\alpha(\Delta)\}$, where $\Delta \in \mathbb{R}$ is a coupling constant and $\alpha$ is a discrete index, such that

$$
\begin{aligned}
{[Q_\alpha(\Delta), Q_\beta(\Delta)]} &= 0, \\
{[Q_\alpha(\Delta), \mathcal{V}]} &= 0, \\
{[Q_\alpha(\Delta), Q_\beta(\Delta')]} &\neq 0,
\end{aligned}
\tag{6}
$$

for all indexes $\alpha, \beta$ and $\forall \Delta, \Delta' \in \mathbb{R}$. Thus each set of commuting charges $\{Q_\alpha(\Delta)\}$ defines a different integrable model, and we find that these models are not superintegrable anymore. The resulting Hamiltonian deformations of the Floquet operators become

$$
\mathcal{V}(\Delta, \lambda) = e^{-iH(\Delta)\lambda} \mathcal{V},
\tag{7}
$$

were $\mathcal{V}$ on the r.h.s. is the original superintegrable circuit.

Intuitively, the existence of different possible deformations is just a consequence of the large degeneracies observed in the superintegrable circuits: it should be possible to split these degeneracies in multiple ways. However, it is still remarkable that this can be performed while conserving integrability. This is the main result of our paper.

Unfortunately we do not have a general mechanism for constructing the different integrable deformations. Instead, we demonstrate the phenomenon on three concrete examples. These examples range from trivial to less trivial and to rather surprising. First we treat the permutation or SWAP circuit, afterwards we add non-trivial phases to the model and show that the phenomenon still exists. Finally we consider the Rule54 model, which is our most involved example.

## 3 The SWAP circuit

This is a rather trivial example. The fundamental two-site unitary is

$$
U_{j,j+1} = \mathcal{P}_{j,j+1},
\tag{8}
$$

where $\mathcal{P}$ is the permutation operator, also called the SWAP gate, which is defined as

$$
\mathcal{P} \mid a, b \rangle = \mid b, a \rangle.
\tag{9}
$$

The time evolution in the resulting quantum circuit is trivial: the Floquet operator $\mathcal{V}$ translates the even (odd) sub-lattice to the left (right) by two sites, respectively. As an effect, the two sub-lattices do not interact with each other at all, and within each sub-lattice of length $L/2$ we just observe a cyclic shift.

The model is clearly superintegrable: Every local operator which acts only on one of the sub-lattices is a glider.

Let us now consider the eigenvalue spectrum of the Floquet operator $\mathcal{V}$. It is useful to introduce the cyclic shifts $\mathcal{U}_R$ and $\mathcal{U}_L$, which perform a shift to the right (left) on the odd (even) sub-lattices. To be more concrete:

$$
\begin{aligned}
\mathcal{U}_R &= \mathcal{P}_{1,3} \mathcal{P}_{3,5} \ldots \mathcal{P}_{L-5,L-3} \mathcal{P}_{L-3,L-1}, \\
\mathcal{U}_L &= \mathcal{P}_{L-2,L} \mathcal{P}_{L-4,L-2} \ldots \mathcal{P}_{4,6} \mathcal{P}_{2,4}.
\end{aligned}
\tag{10}
$$

These operators commute, and we can write

$$
\mathcal{V} = \mathcal{U}_L \mathcal{U}_R.
\tag{11}
$$

The eigenvalues of the cyclic shifts are $e^{\frac{4\pi i J}{L}}$, where $J$ is an integer quantum number. Therefore, the eigenvalues of $\mathcal{V}$ are simply

$$e^{\frac{4\pi i (J_R + J_L)}{L}}, \tag{12}$$

where $J_R$ and $J_L$ are the quantum numbers corresponding to the odd and even sub-lattices. The number of different eigenvalues is $L/2$ for both the odd and the even sub-lattice. Each eigenvalue represents a sector with fixed momentum for both the left and the right translations on the respective sub-lattices, and the dimensions of the sectors are exponentially large in the volume.

In this simple circuit we find an infinite family of gliders: they consist of those local operators that act non-trivially only on one of the sublattices. This gives the idea to construct a practically infinite family of integrable deformations: we can construct an arbitrary integrable model for the two sublattices separately.

In order to have a concrete example, we consider a specific one-parameter family of integrable models. We define

$$H(\Delta) = \sum_{j=1}^{L} h_{j,j+2}(\Delta),$$
$$h_{j,k}(\Delta) = \sigma_j^x \sigma_k^x + \sigma_j^y \sigma_k^y + \Delta \sigma_j^z \sigma_k^z, \tag{13}$$

being the Hamiltonian density of the Heisenberg spin chain, acting on sites $j$ and $k$.

Note that $H(\Delta)$ is a translationally invariant Hamiltonian, which describes two uncoupled XXZ chains on the two sub-lattices. It is straightforward to check that

$$[H(\Delta), \mathcal{V}] = 0, \tag{14}$$

because $\mathcal{V}$ moves the two sub-lattices in an independent way, without altering the states within each sub-lattice. For each $\Delta$ we have a full family of commuting charges, which are those of the XXZ chain, now acting on the sub-lattices. However,

$$[H(\Delta), H(\Delta')] \neq 0, \tag{15}$$

and for each $\Delta$ we have a different integrable model.

This example is indeed rather trivial, but it captures many properties of the more complicated cases.

## 4 The dual unitary phase circuit

Our second example is the SWAP circuit with extra phase factors. In this case we have a real coupling constant $\gamma$ and the two-site gate $U$ is

$$U_{j,j+1} = \exp\left( i\gamma \frac{1 - \sigma_j^z \sigma_{j+1}^z}{2} \right) \mathcal{P}_{j,j+1} = \begin{pmatrix} 1 & & & \\ & & e^{i\gamma} & \\ & e^{i\gamma} & & \\ & & & 1 \end{pmatrix}. \tag{16}$$

The resulting circuit is at the boundary between classical and quantum circuits: If time evolution is started from an initial state which is a tensor product of local basis states (in the given computational basis), then this property is preserved and the resulting dynamics is essentially the same as in permutation circuit. The only difference is that the states acquire various phases

due to the "scattering" of states on the two sub-lattices. These phases do not matter if the initial state is a pure product state, but they do influence the dynamics in the general case when linear combinations are present in the initial state.

The dynamics of the resulting circuit was investigated in [18, 19, 25]. The model is dual unitary [18, 25], which implies that many dynamical properties can be computed without using the traditional integrability properties of the model [19]. However, it is also important that this model emerges from a special limit of the integrable Trotterization of the XXZ model [18, 21].

It is useful to consider the pseudo-energy spectrum of the Floquet operator. We interpret the up spins as the vacuum and the down spins as quasiparticles. In this model the particle numbers are conserved separately for the two sub-lattices. To be more precise, the single step operators $\mathcal{V}_1$ and $\mathcal{V}_2$ exchange the two sub-lattices, but the product $\mathcal{V}$ conserves the particle numbers separately. Therefore, the Hilbert space separates into sectors with fixed particle numbers $N_L$ and $N_R$ in the left-moving and right-moving sub-lattices. The eigenvalues of $\mathcal{V}$ can be determined by the simple observation that

$$\mathcal{V}^{L/2} = \exp\left(i\gamma(L(N_R + N_L) - 4N_L N_R)\right). \tag{17}$$

This is proven easily, by noting that the classical orbits necessarily close after $L/2$ Floquet cycles, and afterwards one has to collect the resulting phase factors. It is clear from this simple formula, that the total number of different eigenvalues of $\mathcal{V}$ can not be bigger than $(L/2)^3$, corresponding to the choice of the root of unity after taking the root of (17), and the different choices for $N_R$ and $N_L$. This implies that almost all the pseudo-energy levels will be again exponentially degenerate.

This model has an infinite family of gliders, therefore it is superintegrable. As simplest gliders let us consider the three-site operators $h_{1,2,3}(\pm\gamma)$ defined as

$$h_{1,2,3}(\gamma) = \begin{pmatrix} 0 & & & \\ & & e^{i\gamma\sigma_2^z} & \\ & e^{-i\gamma\sigma_2^z} & & \\ & & & 0 \end{pmatrix}_{13}. \tag{18}$$

In this mixed representation the matrix indices correspond to the tensor product of spaces 1 and 3, and the matrix elements include the operator $\sigma_2^z$ acting on the second site. Alternatively, we can write

$$h_{1,2,3}(\gamma) = \frac{1}{2}\left[\sin(\gamma)(\sigma_1^x\sigma_3^y - \sigma_1^y\sigma_3^x)\sigma_2^z + \cos(\gamma)(\sigma_1^x\sigma_3^x + \sigma_1^y\sigma_3^y)\right]. \tag{19}$$

These operators propagate ballistically to the right or to the left, depending on their position, given that the sign of the coupling is chosen accordingly. To be precise, we have

$$\mathcal{V}h_{1,2,3}(\gamma)\mathcal{V}^\dagger = h_{3,4,5}(\gamma), \qquad \mathcal{V}h_{2,3,4}(-\gamma)\mathcal{V}^\dagger = h_{0,2,3}(-\gamma). \tag{20}$$

This follows from the formulas

$$U_{12}U_{34}h_{1,2,3}(\gamma)U_{12}^\dagger U_{34}^\dagger = h_{2,3,4}(\gamma),$$
$$U_{12}U_{34}h_{2,3,4}(-\gamma)U_{12}^\dagger U_{34}^\dagger = h_{1,2,3}(-\gamma), \tag{21}$$

which can be checked by direct computation.

Gliders form a closed operator algebra, therefore the squared operators are also gliders, and we get

$$\left(h_{1,2,3}(\gamma)\right)^2 = \left(h_{1,2,3}(-\gamma)\right)^2 = \frac{1 - \sigma_1^z\sigma_3^z}{2}. \tag{22}$$

With this we have two linearly independent local three-site operators, which are both gliders. We show that they lead to a one parameter family of integrable models. We define

$$H(\Delta) = \sum_{j=1}^{L/2} \big( h_{2j,2j+1,2j+2}(\gamma) + h_{2j+1,2j+2,2j+3}(-\gamma) \big) - \Delta \sum_{j=1}^{L} \big( h_{j,j+1,j+2}(\gamma) \big)^2. \tag{23}$$

Note that the first term has a staggering which takes into account the two sub-lattices, but the third term is actually homogeneous, which follows from (22). For $\Delta = 0$ this Hamiltonian appeared in [26] (see also [27]), whereas for general $\Delta$ they are new.

The Hamiltonian is constructed from gliders, therefore

$$[\mathcal{V}, H(\Delta)] = 0, \tag{24}$$

for every $\Delta$. On the other hand

$$[H(\Delta), H(\Delta')] \neq 0. \tag{25}$$

In order to prove our claim of integrable deformations we also need to show that every $H(\Delta)$ defines an integrable spin chain and its higher charges commute with the Floquet operator. This can be proven by embedding the Hamiltonians into a set of commuting transfer matrices and showing that the transfer matrices commute with the update rule $\mathcal{V}$. For this purpose one can use the algebraic framework of [16] developed for medium range spin chains. This method will be used in the next Subsection for the Rule 54 Floquet operator (see the details in Appendix A). However, for the current model we are content with proving the integrability of the commuting Hamiltonian operator $H(\Delta)$ which can be done in a quicker way.

We use the results of the recent work [26]: We show that $H(\Delta)$ can be mapped to a pair of XXZ Hamiltonians acting on the two sub-lattices. To be precise, consider now open boundary conditions, or alternatively an infinite spin chain. We define the diagonal operator

$$\mathcal{D} = \prod_{a<b} e^{i\gamma\sigma_a^z\sigma_b^z/4} \prod_{c>d} e^{-i\gamma\sigma_d^z\sigma_c^z/4}. \tag{26}$$

Here the product over $a$ and $b$ runs over the even sub-lattice, the product over $c$ and $d$ runs over the odd sub-lattice, and it is understood that $1 \leq a < b \leq L$, $1 \leq c < d \leq L$ in the open boundary case. In the infinite volume case the product is to be understood as a formal expression, with no limits for the variables $a, b, c, d$.

It can be shown that

$$\mathcal{D}H(\Delta)\mathcal{D}^{-1} = \sum_j h_{j,j+2}(\Delta), \tag{27}$$

where $h_{j,k}(\Delta)$ is given by (13). Once again, the right hand side describes two uncoupled sub-lattices.

The operator $\mathcal{D}$ is highly non-local, therefore its action makes the two sub-lattices highly entangled. Nevertheless the similarity transformation produces two infinite sets of local conserved charges from those of the original XXZ models. To be more precise, consider the two sets of charges

$$\{Q_\alpha^A(\Delta)\}, \qquad \{Q_\alpha^B(\Delta)\}, \tag{28}$$

where it is understood that they are identical to the charges of the Heisenberg chain with anisotropy $\Delta$, but now acting on the sub-lattices $A$ and $B$, respectively. Let us choose a convention where $\alpha = 2$ corresponds to the Hamiltonian. Then we get from (27)

$$H(\Delta) = \mathcal{D}^{-1}(Q_2^A + Q_2^B)\mathcal{D}, \tag{29}$$

and clearly $H(\Delta)$ will commute with the operators $\mathcal{D}^{-1}Q_\alpha^{A,B}\mathcal{D}$. These latter charges are also local, which follows from the fact the operator densities of $Q_2^A$ and $Q_2^B$ conserve the total magnetization, therefore conjugation with the phase factors dictated by $\mathcal{D}$ can not cause non-local terms. Finally, this proves that for each $\Delta$ we have an infinite family of commuting charges, because we can apply the same similarity transformation for all higher charges.

The alerted reader might notice that the choice of $H(\Delta)$ is somewhat arbitrary. The key relation is (27) which maps the coupled system to two uncoupled integrable spin chains. One could also apply the same similarity transformation to some other integrable model. In this paper we focused on the Heisenberg chain, because it has some similarities with the case of the Rule54 model studied in the next Section.

## 5 The Rule54 model

The Rule54 model was introduced in [5] as a classical cellular automaton on light cone lattices. It is one of the simplest interacting integrable models. The time evolution in the model is as follows.

First we introduce the basis states $|a\rangle$ with $a = 0, 1$ which are identified as the up and down spins, or empty sites and quasiparticles, respectively. Let us also introduce a three site unitary via its action on triple products of basis states:

$$U\Big(|l\rangle \otimes |d\rangle \otimes |r\rangle\Big) = |l\rangle \otimes |u\rangle \otimes |r\rangle\,, \tag{30}$$

where the index $u$ is computed from the indices $l, d, r$ using the equation

$$u = d + l + r + lr\,, \tag{31}$$

which is understood as an equation in the finite field $\mathbb{Z}_2$.

A more conventional notation is as follows. For the basis states we use the alternative notation $|\circ\rangle = |0\rangle$ and $|\bullet\rangle = |1\rangle$ and we also introduce the projectors

$$P^\circ = \frac{1+\sigma^z}{2}\,, \qquad P^\bullet = \frac{1-\sigma^z}{2}\,. \tag{32}$$

With these notations the local three-site unitary can be written as

$$U_{1,2,3} = P_1^\circ P_3^\circ + (1 - P_1^\circ P_3^\circ)\sigma_2^x\,. \tag{33}$$

The Floquet update operation is then constructed as $\mathcal{V} = \mathcal{V}_2 \mathcal{V}_1$ where now

$$\begin{aligned}
\mathcal{V}_1 &= U_{L-1,L,1}\ldots U_{3,4,5} U_{1,2,3}\,, \\
\mathcal{V}_2 &= U_{L,1,2}\ldots U_{4,5,6} U_{2,3,4}\,.
\end{aligned} \tag{34}$$

The physical meaning of the update rule (31) is not transparent from the equation alone, but it becomes clear if one performs simple simulations [5]. One finds that the model describes left-moving and right-moving quasiparticles (solitons) that move with constant speed $\pm 1$, such that the left- and right-movers scatter on each other in a non-trivial way, suffering a displacement of one site in the backwards direction. The expressions for the particle numbers of the right- and left movers are [6, 12]

$$\begin{aligned}
N_R &= \sum_j P_{2j}^\bullet P_{2j+1}^\bullet + P_{2j-1}^\circ P_{2j}^\bullet P_{2j+1}^\circ + P_{2j}^\circ P_{2j+1}^\bullet P_{2j+2}^\circ\,, \\
N_L &= \sum_j P_{2j-1}^\bullet P_{2j}^\bullet + P_{2j-1}^\circ P_{2j}^\bullet P_{2j+1}^\circ + P_{2j}^\circ P_{2j+1}^\bullet P_{2j+2}^\circ\,.
\end{aligned} \tag{35}$$

In the representation given by (34) and (30) the isolated left and right movers are represented by two neighbouring down spins, and a single down spin actually represents a bound state of a left mover and a right mover [12]. This can be read off the formulas in (35).

The model is known to be super-integrable on the classical level: it possesses an exponential number of conservation laws [6]. The conserved quantities correspond to the "particle arrangement" in the left and right moving sectors, which can be recovered in the asymptotic states where left movers and right movers are completely separated [6]. The classical conservation laws can be introduced also in the quantum formulation, in which case the charge densities are simply represented by diagonal operators with the matrix elements being the classical values.

The eigenvalue spectrum of the Floquet operator was treated in [11, 12]. The simplest derivation of the spectrum is through the classical orbits [11]. Let $|\Psi_0\rangle$ stand for an initial state which is a product state in the computational basis. Then the set of states $\{\mathcal{V}^n|\Psi_0\rangle\}$ with $n = 0, 1, 2, \ldots$ form the "classical orbit" of the initial state. It follows from (30) that each one of these states is a product state, therefore it can be interpreted as a classical configuration. The configuration space is finite, and the time evolution is reversible, therefore each orbit is periodic. For each initial state there is a well defined number $n$ which is the smallest non-zero number satisfying

$$\mathcal{V}^n|\Psi_0\rangle = |\Psi_0\rangle. \tag{36}$$

In this case $n$ is called the length of the classical orbit. Eigenstates of the Floquet operator are then found simply as [11]

$$|\Psi_0\rangle + e^{iq}\mathcal{V}|\Psi_0\rangle + e^{2iq}\mathcal{V}^2|\Psi_0\rangle + \cdots + e^{i(n-1)q}\mathcal{V}^{n-1}|\Psi_0\rangle, \tag{37}$$

with some $q$ satisfying $e^{inq} = 1$.

With some abuse of notation, let us now consider an eigenstate with $N_R$ right- and $N_L$ left movers. As it was explained in [11], particle scattering modifies the effective volume available for particle propagation. As an effect, the orbit lengths have to be divisors of

$$\left(\frac{L}{2} + N_L - N_R\right)\left(\frac{L}{2} + N_R - N_L\right). \tag{38}$$

This implies that the eigenvalues of $\mathcal{V}$ are of the form $\lambda_j = e^{i\phi_j}$ with

$$\phi = \frac{2\pi J}{\left(\frac{L}{2} + N_L - N_R\right)\left(\frac{L}{2} + N_R - N_L\right)}, \qquad J \in \mathbb{Z}. \tag{39}$$

Altogether this implies that the maximal number of different eigenvalues can be estimated as $(L/2)^4$. Furthermore, one finds that almost every level will be exponentially degenerate. This degeneracy comes from the various relative placements of the quasiparticles within the sub-lattices, which does not modify the orbit lengths.

## 5.1 Integrable quantum spin chains for the Rule54 model

Now we consider the Hamiltonian deformations of the Rule54 model. The starting point of our investigation is the six-site charge $Q_6$ that was discovered in [12]. It is a conserved charge of the model, and it is a dynamical charge: it generates particle propagation in the model.

The six-site charge can be written as

$$Q_6 = Q_6^+ + Q_6^-, \tag{40}$$

where the two terms are the "chiral" charges given by

$$Q_6^+ = \sum_{j=1}^{L} q^{+6}(j), \qquad Q_6^- = \sum_{j=1}^{L} q^{-6}(j), \tag{41}$$

where $q^{-6}(j) = \left(q^{+6}(j)\right)^\dagger$ and

$$
\begin{aligned}
q^{+6}(1) = q^{+6}_{123456} =\ & P_1^\circ \sigma_2^+ \sigma_3^+ \sigma_4^- \sigma_5^- P_6^\circ + P_2^\circ \sigma_3^+ \sigma_4^- P_5^\circ + P_2^\circ \sigma_3^+ \sigma_4^+ P_5^\bullet P_6^\circ \\
& + P_1^\circ P_2^\bullet \sigma_3^- \sigma_4^- P_5^\circ + P_1^\circ \sigma_2^+ \sigma_3^+ \sigma_4^- P_5^\bullet P_6^\circ + P_1^\bullet P_2^\bullet \sigma_3^+ \sigma_4^- \sigma_5^- P_6^\circ \\
& + P_2^\circ P_3^\bullet \sigma_4^- P_5^\bullet P_6^\bullet + P_1^\bullet P_2^\bullet \sigma_3^+ P_4^\bullet P_5^\circ + P_1^\bullet P_2^\bullet \sigma_3^+ \sigma_4^- P_5^\bullet P_6^\bullet .
\end{aligned}
\tag{42}
$$

We can also see that

$$
q^{-6}_{123456} = q^{+6}_{654321},
\tag{43}
$$

in other words, the chiral charges are space reflections of each other. Our conventions for writing down the charge densities are different from [12], but after summation the resulting charges are identical.

The interpretation of the formula (42) is not evident at first sight. It was explained in [12] that the charge density is in fact a sum over all local processes which preserve the two particle numbers, and which result in a shift of "center of mass" to the right by two units.

The charges above commute with each other and with the update rule [12]:

$$
\left[Q_6^+, Q_6^-\right] = \left[Q_6^\pm, \mathcal{V}\right] = 0.
\tag{44}
$$

Note that (41) describes charges which are translationally invariant. In fact we could also treat charges which are only two-site invariant, but for simplicity we restrict ourselves to the homogeneous charges.

The operator $Q_6$ can be viewed as a Hamiltonian of a spin chain; we will call it the Rule54 Hamiltonian. The coordinate Bethe Ansatz solution of this spin chain was given in [12]. Here we work out the algebraic integrability of this model.

In this work we show that the spin chain defined by $Q_6$ is indeed integrable: it has an infinite family conserved charges, all of which commute with each other and with the time evolution of the Rule54 model. Furthermore, we also introduce a one parameter family of integrable Hamiltonians as follows. We define

$$
H(\Delta) = \sum_{j=1}^{L} h(j|\Delta),
\tag{45}
$$

where $\Delta \in \mathbb{R}$ is again a coupling constant, and $h(j|\Delta)$ is a six-site Hamiltonian density given by

$$
h(j|\Delta) = q^6(j) - \Delta\left(q(j)^6\right)^2, \qquad q^6(j) = q^{+6}(j) + q^{-6}(j).
\tag{46}
$$

The original $Q_6$ defined in [12] corresponds to the choice $\Delta = 0$. We will see that the introduction of the extra term in (46) generates interaction between quasiparticles of the same type. It can be considered as an XXZ-type deformation of the Rule54 Hamiltonian.

We note that

$$
\left(q^6(j)\right)^2 = q^{-6}(j) q^{+6}(j) + q^{+6}(j) q^{-6}(j),
\tag{47}
$$

and this is a projector onto those basis states of a six-site segment of the chain on which $q^6(j)$ acts non-trivially.

In Appendix A we show that $H(\Delta)$ is integrable for every $\Delta$: we construct the corresponding Lax operators and transfer matrices. This is performed using the recently introduced formalism of [16] for medium range spin chains. We also show that the resulting transfer matrices commute with the Floquet operator of the Rule54 model. At the same time, they are truly different integrable models, which is already clear from

$$
[H(\Delta), H(\Delta')] \neq 0.
\tag{48}
$$

This implies that the Hamiltonian deformations

$$\mathcal{V}(\Delta, t) = e^{-iH(\Delta)t}\mathcal{V}, \tag{49}$$

of the Rule54 model are all integrable, but they specify different models for different values of $\Delta$.

The coordinate Bethe Ansatz solution of the model defined by (49) was given in [12] for the case $\Delta = 0$. Here we extend this solution to the case of generic $\Delta$. For simplicity we present here the diagonalization of $H(\Delta)$ in an even volume $L$; the treatment of the combination (49) is straightforward.

In the model there are two types of quasiparticles which we denote by $A$ and $B$. The particle types originate from the left movers and right movers in the original cellular automaton. However, due to the dispersion generated by $H(\Delta)$ the group velocities can be positive and negative for both particle types.

Pseudo-momenta of quasiparticles will be denoted by $p_j^{A,B}$, where $j$ refers to a particle index. Considering Bethe states with $N_A$ and $N_B$ numbers of quasiparticles, we get the Bethe equations

$$
\begin{aligned}
e^{ip_j^A L/2} \prod_{k\neq j} S_{AA}(p_j^A, p_k^A) \prod_k S_{AB}(p_j^A, p_k^B) &= 1, \quad j = 1\ldots N_A, \\
e^{ip_j^B L/2} \prod_{k\neq j} S_{BB}(p_j^B, p_k^B) \prod_k S_{BA}(p_j^B, p_k^A) &= 1, \quad j = 1\ldots N_B.
\end{aligned}
\tag{50}
$$

Here the scattering phases are

$$S_{AA}(p,q) = S_{BB}(p,q) = S^{XXZ}(p,q)e^{-i(p-q)}, \tag{51}$$

$$S_{AB}(p,q) = S_{BA}(p,q) = e^{i(p-q)}, \tag{52}$$

where

$$S^{XXZ}(p,q) = -\frac{e^{ip+iq} - 2\Delta e^{iq} + 1}{e^{ip+iq} - 2\Delta e^{ip} + 1}, \tag{53}$$

is the scattering phase of the XXZ spin chain. We see that the only effect of the $\Delta$ dependent interaction is the modification of $S_{AA}$ and $S_{BB}$. For $\Delta = 0$ we recover the Bethe equations of [12].

The resulting eigenvalues of $H(\Delta)$ are

$$\sum_{j=1}^{N_A} e(p_j^A) + \sum_{j=1}^{N_B} e(p_j^B), \tag{54}$$

with

$$e(p) = 2(\cos(p) - \Delta). \tag{55}$$

The equations (50) can be derived as a generalization of the material presented in [12] For simplicity we do not reproduce the whole computation here. Instead we argue that the Bethe equations follow from the solution of the two-body problem, once the integrability of the model (and thus, factorized scattering) is established [28]. Furthermore, the two-body problem is relatively easily solved, and the additional step required here is just the treatment of the interaction term between quasiparticles of the same type.

The substitution of the concrete formulas (51) into (50) results in equations which can be interpreted as Bethe equations of the XXZ chain in modified volumes. The interpretation of this was already given in [12]: the interaction between quasiparticles modifies the effective space available for particle propagation. This is the same effect which discussed in [29] for the hard rod deformed XXZ models.

## 5.2 The importance of being odd

In the derivation above we assumed that the spin chain length $L$ is even since the Floquet update rule is only defined for even lengths. On the other hand, the spin chain given by the Hamiltonian $H(\Delta)$ is well defined also for odd lengths. Therefore it is interesting and useful to consider the case of odd $L$ as well.

First consider a simpler problem of a Hamiltonian defined as

$$H = \sum_{j=1}^{L} h_{j,j+2}, \tag{56}$$

where $h_{j,j+2}$ is some Hamiltonian density coupling the nearest neighbour sites, and periodic boundary conditions are understood. An example of this appeared in Section 3. If the volume is even, then the chain naturally splits into two uncoupled models on the two sub-lattices. However, if the volume is odd, then the model is equivalent to a single nearest neighbour interacting chain with the same length; this is obtained simply by a reordering of the sites.

Now we consider the Hamiltonian $H(\Delta)$ associated to the Rule54 model. We will see that the odd volumes lead to similar interesting effects, although the mechanism is now more complicated due to the non-trivial interactions.

Let us therefore build the Bethe Ansatz wave functions for $H(\Delta)$ with odd $L$. In this situation we observe an interesting phenomenon: the distinction between the particle types disappears. The particle types $A$ and $B$ originated from the two sub-lattices in the original chain. Now in odd volumes a particle that travels around the volume returns as a particle of the other type. Effectively this means that there is just one particle type in the spectrum, nevertheless the Bethe Ansatz equations can be found by taking a particle around the volume twice, and then collecting all the phase factors. Since we have

$$S_{AA}(p,q)S_{AB}(p,q) = S_{BB}(p,q)S_{BA}(p,q) = S^{XXZ}(p,q), \tag{57}$$

the resulting Bethe equations simplify as

$$e^{ip_j L} \prod_{k \neq j} S^{XXZ}(p_j, p_k) = 1, \quad j = 1 \ldots N. \tag{58}$$

Note that now $L$ is in the exponential phase factor instead of $L/2$ in (50).

We can see that (58) are the Bethe equation of XXZ spin chain with length $L$. Therefore the spectra of the XXZ and the $\Delta$-deformed Rule54 models are the same if the volume is odd. In the particular case of $\Delta = 0$ this also means the spectrum of the model is free in odd volumes, in stark contrast with the volume changing effects in (50). We confirmed these statements with numerical checks in small odd volumes. Thus the model is an example for "free fermions in disguise" [30, 31], although this only works for odd volumes.

A further consequence of this phenomenon is that there exists of a non-local similarity transformation $\mathcal{S}$ for which

$$\mathcal{S}^{-1} H(\Delta) \mathcal{S} = H^{XXZ}(\Delta), \tag{59}$$

when $L$ is odd. At present we do not know what is the form of the transformation, and whether or not it can be constructed using some simple rules. Nevertheless it seems likely that the transformation does not depend on $\Delta$. We leave this question to further research.

Finally we remark, that this situation is somewhat analogous to the one treated in [32], where it was shown that the XXZ spin chain with a special anisotropy has peculiar properties just in the odd length cases. But the mechanism for the "importance of being odd" is different in the two cases.

# 6 Discussion

We showed that for certain selected superintegrable quantum circuits there are families of integrable deformations such that the resulting circuits or spin chains are different integrable models. In other words, the superintegrable models in question do not belong to a single integrable model, instead they lie at the intersection of an (at least) one parameter family of models.

Our most interesting example was the Rule54 model, for which the algebraic reasons for integrability had not been known before. Earlier attempts to embed the model into the standard framework of Yang-Baxter integrability failed. Our current results give a possible explanation for this: there is no single quantum integrable structure behind the Rule54 model, instead there is a one parameter family of Lax operators and $R$-matrices compatible with the model.

Having clarified this issue we are faced with the question: What connects the models that we treated, and how general are our statements about the integrable deformations? At the moment we do not have a complete answer to this question. Nevertheless there are some common points between our examples.

First of all, all our examples are such that the quantum circuit has two sub-lattices and thus two particle types, the left movers and the right movers. The two types of quasiparticles interact with each other (except in the trivial example of the SWAP circuit). On the other hand, quasiparticles of the same type (moving on the same sub-lattice) do not interact with each other, simply because they do not meet (all quasiparticles have the same constant speed). The Hamiltonian deformation drastically changes this picture, because then the quasiparticles get dispersion, and one obtains interaction between quasiparticles of the same type. This was observed and explained in [12], and this is enough to lift the degeneracies of the superintegrable circuit. What we showed here is that there is some freedom in choosing the interactions between quasiparticles of the same type, while still preserving integrability. But all our examples are such that the interaction between quasiparticles of different type are not changed by the deformation; this is indeed completely fixed by the original quantum circuit.

A further common point between our examples is that the original quantum cellular automata are quite easily solved in real space, if we choose to work with product states in the computational basis. In the dual unitary phase circuit this solution was used to generate the non-local similarity transformation $\mathcal{D}$, which uncouples the two sub-lattices (and thus, the two particle types) of the chain. Perhaps a similar uncoupling is possible also in the Rule54 model, using the exact solution. Perhaps this could be performed using effective coordinates, similar to the technique used for hard rod deformed spin chains [29]. In such a case the $\Delta$-dependent Hamiltonian of (46) would arise from two uncoupled Heisenberg chains, using the desired non-local similarity transformation. However, at present these are just vague ideas, and we have not been able to find a concrete formulation of them.

A further way to interpret our results is to consider the discrete time step of the circuit as a symmetry operation for a family of spin chains. This is most natural for the SWAP circuit: Here the operation is the independent translation of the two sub-lattices, which is a symmetry for spin chains where the Hamiltonians and charges are localized on the two sub-lattices separately. In our more complicated examples it is not so evident to interpret the Floquet operator as a symmetry, but in essence we are faced with the same phenomenon.

One of the most interesting questions is, whether these ideas and methods are applicable to other superintegrable models, for example those studied in the recent works [23, 24]. Is particle number conservation crucial for our methods to work? And can we find similar phenomena for those models, where the geometry is slightly different from the brickwork circuit used here? A relevant example could be the cellular automaton of [33], which is perhaps the next simplest model after the Rule54 model, having factorized scattering and three constant

velocities in the classical model (left movers, right movers, and frozen configurations).

We hope to return to these questions in future work.

## Acknowledgments

We are thankful to Aaron Friedman, Sarang Gopalakrishnan, Lorenzo Piroli, Tomaž Prosen, Romain Vasseur and Eric Vernier for inspiring discussions.

## A  Medium range spin chains

Here we summarize the framework developed in [16] which enables us to treat spin chains with medium range interactions. Afterwards we apply these methods to the Hamiltonian deformations of the Rule54 model. The treatment here is just a brief summary, and for a more detailed exposition we refer the reader to the original work [16].

### A.1  General framework

We consider integrable spin chains, where the shortest dynamical charge spans $\alpha \geq 2$ sites. In the case of $\alpha = 2$ we are faced with a standard nearest neighbour interacting Hamiltonian, and in the special case $\alpha = 6$ we obtain the Hamiltonian deformation of the Rule54 model.

Let $V = \mathbb{C}^d$ be the local spaces of the spin chain with some $d$. Our goal is to construct a commuting family of transfer matrices using local Lax operators. In order to obtain an interaction range $\alpha$ we need to consider an auxiliary space $V_A$ which is identical to an $(\alpha-1)$-fold tensor product of the fundamental spaces:

$$V_A = \otimes_{j=1}^{\alpha-1} V_j . \tag{A.1}$$

When writing down space indices we will just use the abbreviation $A = \{1, 2, \ldots, \alpha-1\}$.

The fundamental object is the Lax operator $\mathcal{L}_{A,j}(u)$, which acts on the tensor product of the full auxiliary space $A$ and a single physical space with index $j$. It is also useful to introduced the "checked Lax operator" via

$$\mathcal{L}_{A,j}(u) = \mathcal{P}_{A,j} \check{\mathcal{L}}_{A,j}(u) . \tag{A.2}$$

Here $\mathcal{P}_{A,j}$ is just a short-hand for the following permutation acting on the $\alpha$-fold tensor product $V_A \otimes V_\alpha$:

$$\mathcal{P}_{A,j} = \mathcal{P}_{1,\alpha} \ldots \mathcal{P}_{\alpha-2,\alpha} \mathcal{P}_{\alpha-1,\alpha} , \tag{A.3}$$

where now $\mathcal{P}_{j,k}$ is the two-site permutation operator exchanging the Hilbert spaces at sites $j$ and $k$.

The transfer matrices on a chain of length $L$ are then defined as

$$t(u) = \text{Tr}_A\big[ \mathcal{L}_{A,L}(u) \ldots \mathcal{L}_{A,2}(u) \mathcal{L}_{A,1}(u) \big] . \tag{A.4}$$

The transfer matrices commute for different values of the spectral parameter. This is guaranteed by the so-called RLL relations. Let $R_{A,B}(u,v)$ stand for the so-called $R$-matrix acting on a pair of auxiliary spaces. The RLL relation is

$$R_{A,B}(u,v) \mathcal{L}_{A,j}(u) \mathcal{L}_{B,j}(v) = \mathcal{L}_{B,j}(v) \mathcal{L}_{A,j}(u) R_{A,B}(u,v) . \tag{A.5}$$

It follows from consistency that the $R$-matrix satisfies the Yang-Baxter equations

$$R_{A,B}(\lambda_A, \lambda_B) R_{A,C}(\lambda_A, \lambda_C) R_{B,C}(\lambda_B, \lambda_C) = R_{B,C}(\lambda_B, \lambda_C) R_{A,C}(\lambda_A, \lambda_C) R_{A,B}(\lambda_A, \lambda_B) . \tag{A.6}$$

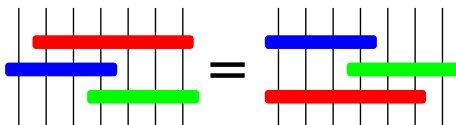

Figure 1: Graphical illustration of the RLL relation for $\alpha = 4$. The red, blue and green boxes are the operators $\check{R}^{(6)}(u,v)$, $\check{L}^{(4)}(u)$ and $\check{L}^{(4)}(v)$.

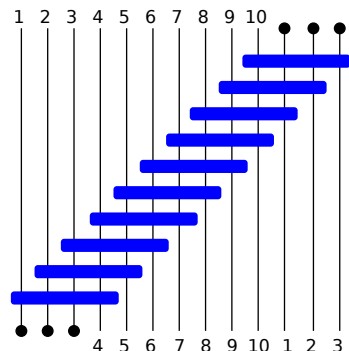

Figure 2: Graphical illustration of the transfer matrix $\check{t}(u)$ for $\alpha = 4$ and $L = 10$. The blue boxes are the operators $\check{L}^{(4)}(u)$. The black dots denotes the summations $\text{Tr}_{L+1,\dots,L+\alpha-1}$.

It can be shown using the standard train argument that if $R_{A,B}(u,v)$ is invertible then

$$[t(u), t(v)] = 0. \tag{A.7}$$

For a later derivation let us show an alternative formulation of the transfer matrix. Let us introduce the R-check operator $\check{R}_{A,B}(u,v) = R_{1,2,\dots,2\alpha-2}(u,v)$ as

$$R_{A,B}(u,v) = \mathcal{P}_{A,B}\check{R}_{A,B}(u,v). \tag{A.8}$$

The RLL-relation reads as

$$\check{R}_2^{(2\alpha-2)}(u,v)\check{\mathcal{L}}_1^{(\alpha)}(u)\check{\mathcal{L}}_\alpha^{(\alpha)}(v) = \check{\mathcal{L}}_1^{(\alpha)}(v)\check{\mathcal{L}}_\alpha^{(\alpha)}(u)\check{R}_1^{(2\alpha-2)}(u,v), \tag{A.9}$$

where we introduced the shorthand notations

$$\check{R}_j^{(2\alpha-2)}(u,v) = \check{R}_{j,j+1,\dots,j+2\alpha-3}(u,v), \tag{A.10}$$

$$\check{\mathcal{L}}_j^{(\alpha)}(u) = \check{\mathcal{L}}_{j,j+1,\dots,j+\alpha-1}(u). \tag{A.11}$$

The graphical illustration of this equation can be seen in figure 1.

We can also introduce a shifted version of the transfer matrix (see figure 2)

$$\check{t}(u) = t(u)\mathcal{U}^{1-\alpha} = \text{Tr}_{L+1,\dots,L+\alpha-1}[\check{\mathcal{L}}_L^{(\alpha)}(u)\dots\check{\mathcal{L}}_1^{(\alpha)}(u)\mathcal{P}_{1,L+1}\dots\mathcal{P}_{\alpha-1,L+\alpha-1}], \tag{A.12}$$

where we also used the shift operator

$$\mathcal{U} = \mathcal{P}_{1,2}\mathcal{P}_{2,3}\dots\mathcal{P}_{L-1,L}. \tag{A.13}$$

Using the RLL-relation we can obtain the following identity

$$\check{R}_{L+1}^{(2\alpha-2)}(u,v)\left[\check{\mathcal{L}}_L^{(\alpha)}(u)\dots\check{\mathcal{L}}_1^{(\alpha)}(u)\right]\left[\check{\mathcal{L}}_{L+\alpha-1}^{(\alpha)}(v)\dots\check{\mathcal{L}}_\alpha^{(\alpha)}(v)\right]$$
$$= \left[\check{\mathcal{L}}_L^{(\alpha)}(v)\dots\check{\mathcal{L}}_1^{(\alpha)}(v)\right]\left[\check{\mathcal{L}}_{L+\alpha-1}^{(\alpha)}(u)\dots\check{\mathcal{L}}_\alpha^{(\alpha)}(u)\right]\check{R}_1^{(2\alpha-2)}(u,v). \tag{A.14}$$

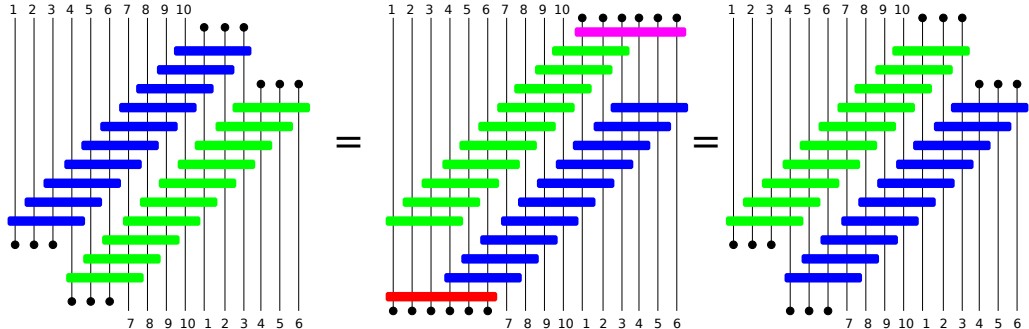

Figure 3: Graphical illustration of the proof of (A.15) for $\alpha = 4$ and $L = 10$. The red, pink, blue and green boxes are the operators $\check{R}^{(6)}(u,v)$, $\check{R}^{(6)}(u,v)^{-1}$, $\check{L}^{(4)}(u)$ and $\check{L}^{(4)}(v)$. The black dots denotes the summations $\text{Tr}_{11,\dots,16}$.

Multiplying this equation with $\check{R}_{L+1}^{(2\alpha-2)}(u,v)^{-1}$ from the left and $\mathcal{P}_{1,L+1}\dots\mathcal{P}_{2\alpha-2,L+2\alpha-2}$ from the right and taking the trace $\text{Tr}_{L+1,\dots,L+\alpha-1}$ we can obtain that

$$\check{t}(u)\mathcal{U}^{\alpha-1}\check{t}(v)\mathcal{U}^{1-\alpha} = \check{t}(v)\mathcal{U}^{\alpha-1}\check{t}(u)\mathcal{U}^{1-\alpha}, \tag{A.15}$$

where we used the following identity

$$\check{R}_1^{(2\alpha-2)}(u,v)\mathcal{P}_{1,L+1}\dots\mathcal{P}_{2\alpha-2,L+2\alpha-2} = \mathcal{P}_{1,L+1}\dots\mathcal{P}_{2\alpha-2,L+2\alpha-2}\check{R}_{L+1}^{(2\alpha-2)}(u,v). \tag{A.16}$$

The graphical proof of (A.15) is illustrated in figure 3 Since

$$\mathcal{U}\check{t}(u) = \check{t}(u)\mathcal{U}, \tag{A.17}$$

the checked transfer matrix also defines commuting quantities

$$[\check{t}(u), \check{t}(v)] = 0. \tag{A.18}$$

From the transfer matrices we can obtain a Hamiltonian with interaction range $\alpha$, if the Lax operator satisfies the initial condition

$$\check{\mathcal{L}}_{A,j}(0) = 1. \tag{A.19}$$

This translates into

$$t(0) = \mathcal{U}^{\alpha-1}. \tag{A.20}$$

The Hamiltonian is then obtained as

$$H = Q_\alpha = \frac{d}{du}\log(t(u))\Big|_{u=0}. \tag{A.21}$$

We get

$$H = \sum_{j=1}^{L} h(j), \tag{A.22}$$

with the Hamiltonian density given by

$$h(j) \equiv h_{j,j+1,\dots,j+\alpha-1} = \frac{d}{du}\check{\mathcal{L}}_{j,j+1,\dots,j+\alpha-1}(u)\Big|_{u=0}. \tag{A.23}$$

## A.2 Application to the Rule54 model

Our first goal is to embed the Rule54 Hamiltonians $H(\Delta)$ into the framework discussed above. For the Lax operator we found

$$\check{\mathcal{L}}_{123456}(u) = A(u) + B(u)h_{123456} + C(u)h_{123456}^2, \tag{A.24}$$

where we defined

$$h_{123456} = \Delta + q_{123456} - \Delta(q_{123456})^2. \tag{A.25}$$

The coefficients are

$$A(u) = \frac{\cosh^2(\eta) - \cosh(u)}{\sinh(u+\eta)\sinh(\eta)}, \qquad B(u) = \frac{\sinh(u)}{\sinh(u+\eta)}, \qquad C(u) = \frac{\cosh(u)-1}{\sinh(u+\eta)\sinh(\eta)}, \tag{A.26}$$

where

$$\Delta = \cosh(\eta). \tag{A.27}$$

In this normalization we have the inversion relation

$$\check{\mathcal{L}}_{123456}(u)\check{\mathcal{L}}_{123456}(-u) = 1. \tag{A.28}$$

This Lax operator satisfies the initial condition (A.19) and its first derivative with respect to $u$ gives the desired operator density of $H(\Delta)$ (apart from an irrelevant constant). We also found the ten-site $R$-matrix which enters the RLL relations (A.5). Unfortunately we did not find a simple functional representation of the $R$-matrix, we obtained it using the program Mathematica as a matrix of size $2^{10} \times 2^{10}$. For general $\Delta$ the concrete matrix is uploaded as a supplementary material with this publication.

The existence of the R-matrix guaranties that the transfer matrices using this Lax operator form a commuting family for every $\Delta$. We also constructed these transfer matrices to confirm this statement. Higher conserved charges can be obtained by taking higher derivatives of the transfer matrices.

We also confirmed that the transfer matrices thus obtained commute with the Floquet operator of the Rule54 model. To be precise, we confirmed the relation

$$[\check{t}(u), \mathcal{V}] = 0. \tag{A.29}$$

The proof is based on the existence of an other seven-site R-matrix for which the following relation is satisfied (see figure 4)

$$\check{R}_3^{(7)}(u)\check{\mathcal{L}}_2^{(6)}(u)\check{\mathcal{L}}_1^{(6)}(u)U_{678}U_{789} = U_{234}U_{345}\check{\mathcal{L}}_4^{(6)}(u)\check{\mathcal{L}}_3^{(6)}(u)\check{R}_1^{(7)}(u). \tag{A.30}$$

Since $[U_{123}, U_{345}] = 0$, we can use the following form of the time step operator (see figure 5)

$$\mathcal{V} = \text{Tr}_{L+1,L+2}\left[U_{L-1}^{(3)}U_L^{(3)}U_{L-3}^{(3)}U_{L-2}^{(3)} \dots U_3^{(3)}U_4^{(3)}U_1^{(3)}U_2^{(3)}\mathcal{P}_{1,L+1}\mathcal{P}_{2,L+2}\right]. \tag{A.31}$$

The proof of (A.29) and (A.18) are the same. We can use the seven-site R-matrix to interchange the operators

$$\check{R}_{L+1}^{(7)}(u)\left[\check{\mathcal{L}}_L^{(6)}(u) \dots \check{\mathcal{L}}_1^{(6)}(u)\right]\left[U_{L+4}^{(3)}U_{L+5}^{(3)} \dots U_6^{(3)}U_7^{(3)}\right]$$
$$= \left[U_L^{(3)}U_{L+1}^{(3)} \dots U_2^{(3)}U_3^{(3)}\right]\left[\check{\mathcal{L}}_{L+2}^{(6)}(u) \dots \check{\mathcal{L}}_3^{(6)}(u)\right]\check{R}_1^{(7)}(u), \tag{A.32}$$

where we also used that

$$[U_{123}, \check{\mathcal{L}}_{345678}(u)] = 0. \tag{A.33}$$

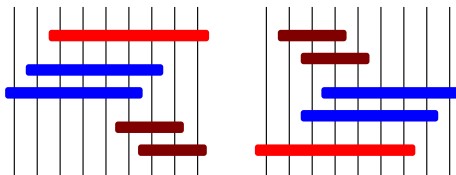

Figure 4: Graphical illustration of the relation (A.30). The red, blue and burgundy boxes are the operators $\check{R}^{(7)}(u,v)$, $\check{L}^{(6)}(u)$ and $U^{(3)}$.

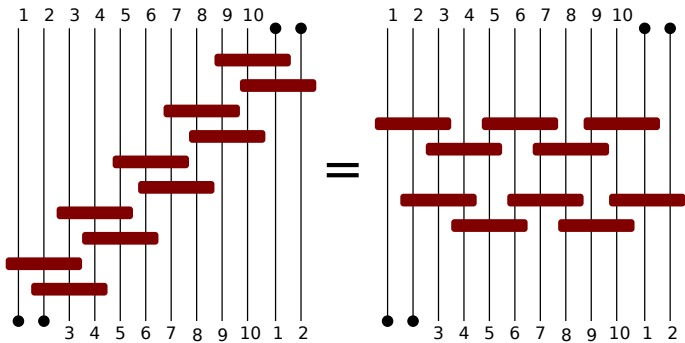

Figure 5: Graphical illustration of the time step operator $\mathcal{V}$ for $L = 10$. The burgundy boxes are the operators $U^{(3)}$. The black dots denotes the summations $\text{Tr}_{11,12}$.

This equation leads us to

$$\check{t}(u)\mathcal{U}^5\mathcal{V}\mathcal{U}^{-5} = \mathcal{U}\mathcal{V}\mathcal{U}^{-1}\mathcal{U}^4\check{t}(u)\mathcal{U}^{-4}. \tag{A.34}$$

Since

$$\mathcal{U}^2\mathcal{V} = \mathcal{V}\mathcal{U}^2, \qquad \mathcal{U}\check{t}(u) = \check{t}(u)\mathcal{U}, \tag{A.35}$$

we just proved (A.29). The graphical illustration of proof is in figure 6.

Interestingly, the Floquet operator itself is not reproduced directly by $t(u)$, neither for a special value of $u$, nor for special limits.

We note that the functional form (A.24) is essentially the same as in the XXZ Heisenberg spin chain, although this is not clear just from (A.24). For completeness we explain the connection. Let us start with the Hamiltonian density of the XX chain:

$$h_{12} = \sigma_1^+\sigma_2^- + \sigma_2^+\sigma_1^-. \tag{A.36}$$

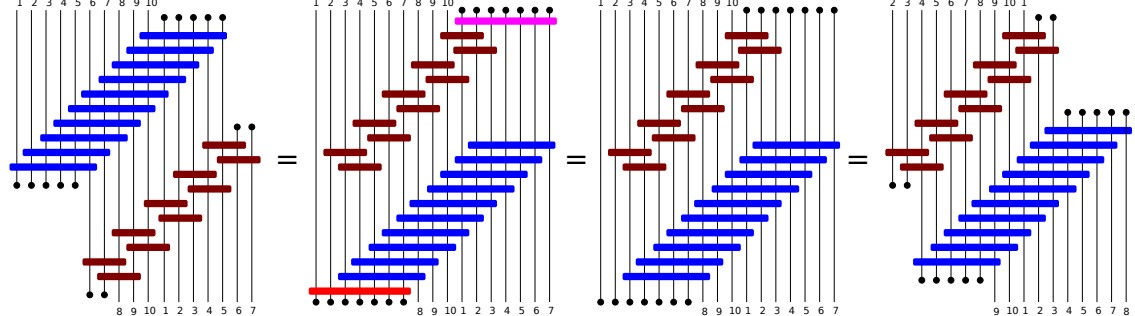

Figure 6: Graphical illustration of the proof of (A.29). The red, pink, blue and burgundy boxes are the operators $\check{R}^{(7)}(u,v)$, $\check{R}^{(7)}(u,v)^{-1}$, $\check{L}^{(6)}(u)$ and $U^{(3)}$. The black dots denotes the summations $\text{Tr}_{11,\dots,17}$.

The Hamiltonian density of the XXZ chain with anisotropy $\Delta$ can be expressed as

$$h_{12}(\Delta) = \Delta + h_{12} - \Delta(h_{12})^2. \tag{A.37}$$

The Lax operator reads as

$$\check{\mathcal{L}}_{12}(u) = A(u) + B(u)h_{12} + C(u)h_{12}^2. \tag{A.38}$$

Using the explicit forms of the coefficients $A(u), B(u), C(u)$ and the operator $h_{12}(\Delta)$ we found the usual XXZ Lax operator

$$\check{\mathcal{L}}_{12}(u) = \frac{1}{\sinh(u+\eta)} \begin{pmatrix} \sinh(u+\eta) & 0 & 0 & 0 \\ 0 & \sinh(\eta) & \sinh(u) & 0 \\ 0 & \sinh(u) & \sinh(\eta) & 0 \\ 0 & 0 & 0 & \sinh(u+\eta) \end{pmatrix}. \tag{A.39}$$

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
