# Peer review of "Integrable deformations of superintegrable quantum circuits"

_SciPost Physics, doi:SciPost Phys. 16, 114 (2024)_

## Round 4 · Referee Report · Anonymous (Referee 1) · 2024-1-22

Strengths

1- Solves an outstanding problem of the integrability structure behind "Rule 54". 2- Clearly written and accessible.

Report

The manuscript discusses integrable deformations of "superintegrable" quantum circuits: They superimpose an integrable Hamiltonian to a superintegrable circuit so that the resulting model no-longer exhibits an exponential number of conserved quantities. They show that the integrable Hamiltonian (and all its higher conserved charges) commute with the original model, which implies that the new model is also integrable in the usual sense. They consider three examples, a circuit made of swap gates, a circuit of swap gates with phases, and the so-called "Rule 54" circuit. The latter example is the most interesting (and nontrivial) of the three.

The result on the integrability of Rule 54 represents a fundamental breakthrough: the model has been observed to be "solvable" in many different contexts, but until now the algebraic reasons for its integrability had not been understood. This paper answers the question by exhibiting a family of integrable models, whose transfer matrices commute with the unitary operator, and thus positions the model at an intersection of a (at least one) family of Yang-Baxter integrable models. Moreover, the contents of the paper are clearly and sensibly structured, which makes the manuscript easy to follow and accessible. Therefore I recommend it for publication in SciPost Physics.

Requested changes

I have a few comments/questions for the authors.

1- Paragraph before (4.13): "therefore its action makes the two sublattices highly entangled, both in equilibrium, and non-equilibrium situations."

I find this sentence a bit confusing - what do "equilibrium and non-equilibrium situations" refer to? The way I understand it, if one starts with a state that is a product state with respect to the two sublattices, and applies the operator $\mathcal{D}$ on it, the resulting state is entangled between the two sublattices. But this is a general statement for states, I don't see how this relates to either equilibrium or non-equilibrium.

2- Section 4: How do we know that all the higher charges also commute with $\mathcal{V}$? From the discussion it is clear that $H(\Delta)$ commutes with $\mathcal{V}$, and that it admits a family of commuting charges. However, it is not obvious to me that all these additional charges commute with $\mathcal{V}$.

3- Last few paragraphs of Sec. 5.2: Why are there so many exclamation marks? I can understand the excitement of the authors, but in my personal opinion some of the exclamation marks could be replaced by fullstops.

4- A common point of all the three examples is the fact that "evolution in space" can be formulated as a valid local dynamical map. Do the authors expect that this plays any role? A related question is, whether the deterministic map of Ref. [17] could be treated in an analogous way. Could the treatment be extended to that case? I would naively expect its integrable structure to be somehow related. I understand that this might not be necessarily a straightforward question to answer.

---

## Round 4 · Referee Report · Anonymous (Referee 2) · 2024-3-11

Strengths

  1. This paper makes substantial progress extending the method initially applied to Rule54 in Ref. 12 to a variety of superintegrable quantum spin chains.
  2. The paper provides new insights into an intriguing class of model, relevant to a surprisingly large swath of 1d systems, as well as topical areas of quantum integrability vs chaos, quantum cellular automata, and connections between the two.
  3. The authors discover that there are multiple ways to break superintegrability to integrability, finding continuous families of integrable deformations, which is neat.
  4. The authors find a new way to deform Rule54 compared to Ref. 12 that they connect to the physics of XXZ in an interesting way.

Weaknesses

The only weakness that stands out to me is that no general framework was identified for describing what makes models superintegrable and an exhaustive classification of integrable deformations thereof. However, given that this might not even be possible, it is not a reasonable criticism of this submission.

Report

I recommend this paper for publication in SciPost. It meets all general acceptance criteria and the 2nd and/or 3rd expectations for the journal. While there are a number of grammatical errors, the paper is quite easy to read, especially compared to other papers on integrability, and I commend the authors for their presentation and results. I have only some minor comments / suggestions; see below.

Requested changes

  1. Expand "Intro" to "Introduction."
  2. The term "completely integrable" does not seem to be a standard term? If it is not a standard term, I strongly recommend changing it to "standard integrability" or just saying "integrable" because it's weird to have something (superintegrability) that is more than "complete."
  3. As a minor clarification on the Kepler problem mentioned in the intro, how many conserved charges are there for $n=2d$ degrees of freedom? I got distracted by this point and started thinking about it, so might be good to give that information.
  4. In the start of the 3rd paragraph of the intro, what's "less clear" about regular integrability in quantum systems? Is the point just that there isn't a precise notion that you need at least $n+1$ conserved charges? If so, it would help to state this clearly as part of the first sentence.
  5. In the 5th intro paragraph, when mentioning "particles (solitons)" it should probably be "quasiparticles (solitons)" instead. Also, should state that the number of local conservation laws for Rule54 is exponential "in the number of spins."
  6. In general, I suggest reading through again to ensure consistent formatting, i.e., should always say "six-site" (with the number spelled out and a hyphen), but there is a mix of conventions currently. There are other similar inconsistencies as well. While they don't detract from the point, the authors may wish to polish the text a bit before publication.
  7. At the end of the 6th intro paragraph, replace "original classical model" with "superintegrable model" maybe?
  8. Also in that paragraph and elsewhere, I think it's wrong to say that Ref. 12 failed to clarify the algebraic structure of Rule54, as this was not the intent of that paper, which was merely to identify a deformation of Rule54 that could be solved via coordinate Bethe Ansatz, and work out the basic TBA and GHD properties.
  9. In the paragraph beginning with "We find a somewhat unexpected phenomenon..." maybe clarify whether the deformation is never unique, or just sometimes not unique (at least among the examples considered). And when it's mentioned that there exists a "one-parameter family of Hamiltonians" is this for all superintegrable cellular automata or just Rule54?
  10. There are a number of places where it's not clear whether the authors refer to all (superintegrable) cellular automata or only particular ones. For example, the last sentence of the intro.
  11. First sentence of Sec. 2, maybe clarify "both continuous (i.e., Hamiltonian) and discrete (i.e., Floquet) time evolution"
  12. Before or after Eq. 2.1, clarify that this structure defines the notion of a brickwork circuit.
  13. After Eq. 2.2, clarify that translation invariance of some kind is assumed, because this is not clear from Eq. 2.2 but is assumed in the sentence that follows. Maybe just say all of the multi-spin gates are the same?
  14. In the sentence that immediately follows, note that there are more options than chaos and integrability, including localization, fragmentation / shattering, scars, etc.
  15. After the bullet points, I did not understand the phrase "the range of the operator density of the charges..." maybe just say "the size of the charge's support" or the "number of consecutive sites acted upon by a charge operator" or otherwise clarify the notion of charge operator size? The same should be clarified (i.e., the meaning of "range") in the italicized statement. Authors could consider using AMS Theorem definition environment, but it's also clear as is.
  16. In the paragraph beginning "There is a further common characteristic..." I would replace "generic integrable systems" with "standard integrable systems," and the second sentence requires a little more clarification of the nature of the degeneracies, both in the superintegrable and standard integrable cases. Not much more, but it's vague as written. A citation could help.
  17. In that same paragraph, why mention ground states at all? Integrability applies to the entire spectrum, and as noted, this doesn't matter for circuits. Also replace "up to a shift by $2\pi$" with "modulo $2 \pi$"
  18. Is the statement below Eq. 2.4 always true or sometimes true?
  19. Remove "It is clear" from the next sentence. I generally advise against saying things like "clearly" or "obviously" because if true, it need not be said.
  20. I would remove the heading for "Sec. 2.1" and just absorb into Sec. 2.
  21. In Eq. 2.5 and below, I would replace $t$ with $\lambda$ or similar to avoid confusion between the discrete time $t$ and the continuous deformation parameter.
  22. I would replace Eqs. 2.6--2.8 with the subequation environment, and also specify, e.g., $\forall \alpha, \beta \in \mathbb{Z}$ and $\forall \Delta \in \mathbb{R}$.
  23. Is there a relation between $\Delta$ and $\lambda$ or are these independent continuous deformation parameters?
  24. Last paragraph of Sec. 2.1, clarify no general mechanism for constructing different integrable deformations "other than Eq. 2.5" if correct. Consider deleting last two sentences, as they are unnecessary.
  25. Need to define the permutation operator, e.g., $\mathcal{P} \left| a , b \right\rangle = \left| b , a \right\rangle ~,~~ \forall a,b$. Also, Eq. 3.2 is redundant to Eq. 2.2, and Eq. 3.3 to Eq. 2.1. Just cross reference back to those equations.
  26. The sentence below Eq. 3.3 seems to suggest that $J$ is an integer, and since $L$ is also an integer, $e^{4 \pi i J L}=1$. So $J$ should be of the form $n/2L$ for integer $n$, no?
  27. Combine Eqs. 3.5 and 3.6 into one line. Specify two-site translation invariance above Eq. 3.7, and replace with "obvious that" with "straightforward to check that" or similar.
  28. Should also clarify that / whether Eq. 3.5 is a charge of the integrable model $\mathcal{V}$ for this SWAP circuit. Maybe a comment on other families of charges other than Eq. 3.5?
  29. Skipping ahead, between Eq. 5.15 and 5.16, rephrase comment about Ref. 12 to say that "here we work out the ABA for Rule54" instead?
  30. Are there no $\pm$s in Eq. 5.17? The new deformation is cool.
  31. Finally, I think some of the content of appendices may be reasonably moved to the main text. I did not read these as carefully, and I leave this to the authors' discretion.

---

## Round 5 · Referee Report · Anonymous (Referee 2) · 2024-3-28

Strengths

See prior report

Weaknesses

N/A

Report

The authors have sufficiently addressed my comments from my previous report. I recommend the submission for publication.

Requested changes

N/A

---

## Round 5 · Referee Report · Anonymous (Referee 1) · 2024-4-1

Report

The authors have satisfactorily answered all my minor questions, and those of the second referee, therefore I recommend the paper for publication.

---

## Round 5 · Author Response

We are thankful to the Referees for the careful reading of the text and for the various comments and requests. Here are our replies.

Referee 1

  1. Indeed this was not meaningful, we deleted it.

  2. The referee is right that the commutation of the additional charges with the Floquet operator does not follow from our argument in Section 4. For the model of Section 4, We are content to prove that the Floquet operator commutes with an integrable Hamiltonian. For Rule 54, we have proved precisely that the additional charges commutate with the update rule, and this method could be used in section 4, but it would require more modifications. Based on the above, we have modified the paragraph following Equation 4.10.

  3. We deleted the exclamation marks. Indeed we were very excited, because this was unexpected, and whenever we tell this to other researchers they are also surprised and they like this peculiar behaviour. Nevertheless it is indeed not necessary to keep so many exclamation marks.

  4. This is a good question. At this moment we don't know, whether the two properties (integrable deformations of superintegrable circuits, and the possibility to define the evolution in space) are related. They might be. Also, we think that the evolution in space of [17] is likely related, but we have not worked on it, and this would require longer computations than what is reasonably expected from a minor revision.

Referee 2

  1. Done.
  2. We deleted the word completely'' and kept onlyintegrable'' and ``integrability''.
  3. We added a footnote on page 1. summerizing ing the superintegrability of the Kepler problem.
  4. We added a footnote on page 2. explaining the key difference between the classical an the quantum case.
  5. We performed the required modifications.
  6. We applied a unified convention, now it is indeed ``six-site'', etc.
  7. We exchanged "original classical model" to "original superintegrable model".
  8. In the Introduction we say that ``However, the algebraic integrability of the resulting model was not clarified in [12].'' We still think that this is a true statement. With this sentence we do not say that the authors of [12] failed to solve the problem, we do not say that they attempted to solve the problem. But we want to say that the problem exists, and in that work it was simply not solved. So here we did not make any changes, and we hope the referee can agree with this.
  9. We tried to clarify the paragraph starting with ``We find a somewhat unexpected phenomenon: '' We can not say anything about all superintegrable cellular automata, but we are claiming the statements about the examples that we consider in the work.
  10. We added a clarification to the very last sentence of the Introduction.
  11. We clarified this.
  12. We added "and the structure of $\VV$ defines the notion of a brickwork circuit." to the sentence after (2.2).
  13. We added an extra explanation.
  14. We added also some other dynamical possibilities.
  15. We added a footnote about this on page 3.
  16. We replaced generic'' withstandard'', however, following this it is not clear to us what the referee requests.
  17. We put here modulo 2pi''. But afterwards, we don't understand the comment of the referee. Our sentence starts with Whereas the concept of a ground state is missing in such models''. We are just explaining that the level spacing statistics can be defined even in this case, when there is no natural beginning'' andend'' of the spectrum.
  18. We clarified that statement. The existence of one glider is in fact enough to guarantee exponential increase of the number of conserved charges, in the limit of large sizes.
  19. We deleted it.
  20. We deleted it.
  21. We replaced this.
  22. We performed the requested changes.
  23. We added a new equation (2.7) clarifying this.
  24. It is not clear to us which sentences should we delete. The very last two sentences of Section 2 are explaining the plan of what we intend to do. It is not clear why we should delete those.
  25. We added the definition of the permutation operator.
  26. We corrected this.
  27. We replaced "obvious that" with "straightforward to check that", but it is not clear why should we put (3.5) and (3.6) into the same equation.
  28. We added an explanatory paragraph before eq. (3.6).
  29. We changed this.
  30. We added the signs.
  31. We still think that the Appendices should be presented separately.

---

## Editorial Decision

published